# Inflammation, Mitochondria and Natural Compounds Together in the Circle of Trust

**DOI:** 10.3390/ijms24076106

**Published:** 2023-03-24

**Authors:** Salvatore Nesci, Anna Spagnoletta, Francesca Oppedisano

**Affiliations:** 1Department of Veterinary Medical Sciences, Alma Mater Studiorum-Università di Bologna, 40064 Ozzano Emilia, Italy; salvatore.nesci@unibo.it; 2ENEA Italian National Agency for New Technologies, Energy and Sustainable Economic Development, Trisaia Research Center, 75026 Rotondella, Italy; 3Department of Health Sciences, Institute of Research for Food Safety and Health (IRC-FSH), University “Magna Graecia” of Catanzaro, 88100 Catanzaro, Italy; oppedisanof@libero.it

**Keywords:** mitochondrial dysfunction, reactive oxygen species, inflammatory response, cytokines, nutraceuticals, phytosome

## Abstract

Human diseases are characterized by the perpetuation of an inflammatory condition in which the levels of Reactive Oxygen Species (ROS) are quite high. Excessive ROS production leads to DNA damage, protein carbonylation and lipid peroxidation, conditions that lead to a worsening of inflammatory disorders. In particular, compromised mitochondria sustain a stressful condition in the cell, such that mitochondrial dysfunctions become pathogenic, causing human disorders related to inflammatory reactions. Indeed, the triggered inflammation loses its beneficial properties and turns harmful if dysregulation and dysfunctions are not addressed. Thus, reducing oxidative stress with ROS scavenger compounds has proven to be a successful approach to reducing inflammation. Among these, natural compounds, in particular, polyphenols, alkaloids and coenzyme Q_10_, thanks to their antioxidant properties, are capable of inhibiting the activation of NF-κB and the expression of target genes, including those involved in inflammation. Even more, clinical trials, and in vivo and in vitro studies have demonstrated the antioxidant and anti-inflammatory effects of phytosomes, which are capable of increasing the bioavailability and effectiveness of natural compounds, and have long been considered an effective non-pharmacological therapy. Therefore, in this review, we wanted to highlight the relationship between inflammation, altered mitochondrial oxidative activity in pathological conditions, and the beneficial effects of phytosomes. To this end, a PubMed literature search was conducted with a focus on various in vitro and in vivo studies and clinical trials from 2014 to 2022.

## 1. Introduction

Complex disorders in human diseases are characterized by a cycle of inflammatory perpetuation. The “druggability” of target components of the inflammatory process represents a novel and exciting strategy for therapeutic interventions aimed at treating chronic diseases and beyond [1,2]. Reactive oxygen species (ROS) levels are typically rather high in inflammatory disorders. Increased ROS generation leads to DNA damage, protein carbonylation, and lipid peroxidation, which worsen inflammatory disorders. Impaired mitochondria are the main players in sustaining stress condition in the cell [3]. Mitochondrial dysfunction is pathogenic, causing human disorders related to inflammatory reactions [4]. The flow of energy through an organic system has led to the development of the biological complexity of living things, where energy-producing processes are established to take place in the mitochondria. As a consequence of bioenergetic failure, this perspective on human diseases provides a pathophysiological and molecular mechanism for neuromuscular and neurodegenerative disease, metabolic non-communicable and communicable diseases, autoimmune diseases, ageing, and cancer, which deeply affect the mitochondria. [5]. Alterations in cellular and organismal homeostasis are caused by a variety of chemical and physical factors of cellular damage. However, when these dysregulation and dysfunctions are not countered, the inflammation triggered ceases to be useful and becomes harmful [6].

Reducing oxidative stress with ROS scavenger compounds has proven to be a successful approach for reducing inflammation. Indeed, emerging therapies for the treatment of inflammatory illnesses include electron transfer, based on antioxidant compounds. Polyphenols, which include phenolic acids, stilbenes, flavonoids (flavonols, flavanols, anthocyanins, flavanones, flavones, and isoflavones), curcuminoids, carotenoids, capsaicinoids and capsinoids, isothiocyanates, catechins, and vitamins, are significant classes of compounds with antioxidant properties, found in plants. The ROS scavenger activities of these compounds, their ability to inhibit NF-κB (nuclear factor κ-light-chain-enhancer of activated B cells) activation, and their ability to suppress the expression of target genes, including those involved in inflammation, are determined by their antioxidant activities [7]. In this review, we highlight the relationships between inflammation, the altered mitochondrial oxidative activity in pathological conditions and the beneficial effects of nutraceutical compounds.

## 2. Inflammation: An Overview

The cellular biological response against aggression, caused by infectious and non-infectious agents (Gram-positive and Gram-negative bacteria, viruses, parasites, toxic substances, or irradiation), is defined as inflammation [2]. Therefore, as a result of an inflammatory process, there is a disruption of or damage to homeostatic processes at the cellular level. The inflammatory process can occur with acute and/or chronic inflammatory responses, thus, involving different organs (heart, liver, kidneys, brain, pancreas, lungs, intestines, etc.). All these events can potentially lead to tissue damage or disease, such as atherosclerosis, Type 2 diabetes, obesity, neurodegenerative diseases, dysbiosis and cancer [2,8].

However, the inflammatory process, if it occurs in a timely and complete manner, is of paramount importance, because it ensures the survival of the organism during an infection or injury [9].

Its main goals are to neutralize harmful stimuli, allow continued homeostasis of damaged tissues, and initiate the healing process [10].

At the tissue level, inflammation is characterized by five cardinal signs: redness, swelling, heat, pain, and tissue dysfunction (*rubor, tumor, calor, dolor et functio laesa*). The first four signs were clinically defined in the first century AD by a Roman physician, Cornelius Celsus. The fifth was added centuries later by Rudolph Virchow in 1858 [2].

Such signs then are nothing more than the macroscopic result of the local responses that occur during the inflammatory process (changes in vascular permeability, leukocyte recruitment and accumulation, and the release of inflammatory mediators) as a result of the triggering of a complex and coordinated communication between the different cells of the immune system (neutrophils, monocytes and lymphocytes) and the blood vessels [10].

Regardless of the nature of the stimulus and location in the body, different inflammatory processes all share common mechanisms, which have been extensively discussed and explored in the literature [11], and which we have attempted to summarize in the text and Figure 1.

### 2.1. Recognition of Damaging Stimuli by Cell Surface Receptors: PRRs (Pattern Recognition Receptors)

PRRs are proteins expressed and present both on the plasma membrane and in the cytoplasm of macrophages, neutrophils, eosinophils, mast cells, natural killer (NK) cells, dendritic cells, and also in other cells (e.g., endothelial cells). They play a crucial role in the main functions of the innate immune system. PRRs are germline-encoded host sensors, which detect molecules that are typical for pathogens [12].

They recognize two classes of molecules/molecular patterns:Microbial structures, known as Pathogen Associated Molecular Patterns (PAMPs), expressed by pathogenic microbes [13];Biomolecules, known as Damage-Associated Molecular Patterns (DAMPs), expressed by host cells during cell/tissue damage or cell death [14].

These molecules also belong to the group of endogenous stress signals (e.g., uric acid, extracellular ATP) that can initiate and perpetuate a noninfectious type of inflammatory response, called “sterile inflammation” [15].

Families of PRRs include Toll-like receptors (TLRs), C-type lectin receptors (CLRs), retinoic acid-inducible gene (RIG)-I-like receptors (RLRs), and NOD-like receptors (NLRs) [16]. PRRs also mediate the initiation of the antigen-specific adaptive immune response and the release of Inflammatory Interleukins (ILs) [17,18].

### 2.2. Activation of Inflammatory Pathways

Inflammatory pathways primarily involve inflammatory stimuli, which are used for the activation of intracellular signaling pathways, inflammatory mediators, and all common regulatory pathways that are composed of an intricate cascade of molecular signals.

Therefore, ILs such as interleukin-1β (IL-1β), interleukin-6 (IL-6), and tumor necrosis factor-α (TNF-α), as well as microbial products, act through receptor interactions with TLRs, IL-1 receptor (IL-1R), IL-6 receptor (IL-6R), and TNF receptor (TNFR) [19].

Receptor activation triggers important intracellular signaling pathways, including:The mitogen-activated protein kinase (MAPK) pathways [20] and the NF-κB pathways [21].The Janus Kinase (JAK) transducer and Signal Transducer and Activator of Transcription (STAT) signaling pathways [22].The Hypoxia-Inducible Factors-1 Alpha (HIF-1α) pathway [23].

### 2.3. Activation and Release of Inflammatory Markers

Activation by pro-inflammatory stimuli of immune and inflammatory cell origin, such as macrophages, monocytes, lymphocytes and adipocytes, induces the production of inflammatory ILs, inflammatory proteins and enzymes [24].

Specifically, the inflammatory cytokines involved are classified as IL (IL-1β, IL-6), CSF, IFN, TNF-α, TGF, and chemokines. These cytokines are mainly produced by cells present at the site of infection or injury, and are responsible for leukocyte recruitment [25].

The concerted action between pro- and anti-inflammatory cytokines (IL-4, IL-6, IL-10, IL-11, and IL-13) [26] ensures the proper modulation of the immune response in the event of an infection or inflammation. Excessive production of inflammatory cytokines can lead to tissue damage, hemodynamic changes, organ failure, and, ultimately, death [27].

Inflammation and metabolism are closely interconnected. Many studies have shown that cellular oxidative stress can also be an important pro-inflammatory stimulus, as it can increase the gene expression of growth factors, inflammatory cytokines, and chemokines. In particular, it has been shown that high levels of oxidative stress can induce the production of ROS, malondialdehyde (MDA), 8-hydroxy-2-deoxyguanosine (8-OHdG) and isoprostanes, each of which can activate various transcription factors, including NF-κB, Activator Protein-1, p53 and STAT [28,29].

The cellular compartments that are most involved in ROS production are the cytoplasm, peroxisome, endoplasmic reticulum, and mitochondria [30].

The latter undoubtedly represent the primary source of ROS (mitochondrial ROS, mtROS) and are a major signalling hub at the cellular level [31].

Mitochondrial dysfunction and associated excessive levels of mtROS are major supporters of inflammation [32] that drive the pathogenesis of many diseases, including neurodegenerative disorders, cancer, viral and bacterial infections, cardiovascular diseases, metabolic syndromes, and autoimmune disorders [31].

Partegnani et al. define this as “mito-inflammation”, a new concept that identifies the compartmentalization of the inflammatory process, in which the mitochondrion acts as a central regulator, checkpoint, and arbiter. This new view of the mitochondrion as a key player in inflammation represents a major advancement, in both the increased understanding of the various mechanisms of mtROS communication, as well as in the study and development of new mitochondrial anti-inflammatory therapies [33].

### 2.4. Recruitment of Inflammatory Cells

All cells involved in the inflammatory process are recruited to sites of tissue injury by factors released by damaged epithelial and endothelial cells, and trigger an inflammatory cascade, along with chemokines and growth factors. The first cells attracted to an injury site are neutrophils, followed by monocytes/macrophages, dendritic cells, lymphocytes, NK cells, T cells, B cells), and mast cells [34].

Neutrophils and monocytes/macrophages represent the first cells of the innate immune response and, therefore, are critical for the initiation, maintenance, and resolution of inflammation [35].

### 2.5. Resolution of Inflammation

The process of resolution of inflammation is of fundamental importance, because it serves to limit both the development of further tissue damage and the onset of persistent chronic inflammation. The presence of mediators (lipoxin, resolvins, protectin, etc.) [36], even at this final stage of the inflammatory process, plays a key role, since their presence drives, both locally and systemically, a process of reduction in chemokine gradients over time.

This process leads to a gradual restoration of tissue homeostasis (reduction or cessation of tissue infiltration by neutrophils, apoptosis of exhausted neutrophils), to a counter-regulation of chemokines and cytokines, and to the initiation of healing [1].

Dysregulation of these processes can lead to uncontrolled chronic inflammation, resulting in the development of several associated diseases such as cardiovascular disease, atherosclerosis, Type 2 diabetes, rheumatoid arthritis, and cancers [37,38].

## 3. Mitochondrial Bioenergetics and Dysfunction

The mitochondria have always been considered the powerhouse of the cell. During aerobic metabolism, mitochondria are the primary source of ATP production. The chemiosmotic hypothesis explains the bioenergetic concept that governs the Oxidative Phosphorylation (OXPHOS) system in mitochondria [39,40] (Figure 2). OXPHOS activity relies on the inner mitochondrial membrane (IMM)-embedded enzyme (super)complexes [41]. As a key mechanism of energy production in cells, they are composed of respiratory machinery, which is responsible for substrate oxidation, working in concert towards the goal of producing ATP by the way of ADP phosphorylation. Mitochondrial respiration is sustained by the electron transfer chain (ETC) that drives the flow of electrons to the final acceptor, oxygen (O_2_), by four respiratory complexes. These are named as follows: NADH:ubiquinone oxidoreductase, or Complex I; succinate dehydrogenase, or Complex II; cytochrome *bc*_1_ oxidoreductase, or dimeric Complex III_2_; cytochrome *c* oxidase, or Complex IV. Moreover, to sustain the electron flow from NADH-dependent Complex I oxidation, or succinate-dependent Complex II oxidation, to O_2_-dependent Complex IV reduction, two mobile electron carriers complete the ETC, i.e., the membrane-embedded hydrophobic coenzyme Q_10_ (CoQ_10_) and the soluble cytochrome *c*. CoQ_10_ is the electron shuttle between either Complex I or Complex II and Complex III_2_, whereas cytochrome *c* facilitates the exchange of electrons from Complex III_2_ to Complex IV [42].

The redox energy of electron transfer is coupled to H^+^ translocation during respiration, building a proton motive force (Δ*p*) that drives ATP synthesis. Complex V, known as ATP synthase, uses the Δ*p* created by ETC to phosphorylate ADP into ATP, the universal energy unit of cells. However, the enzyme can also work in the opposite direction when the Δ*p* drops under phato (physio) logical conditions, and it exploits the energy of the phosphoanhydride bond of the ATP molecules to re-energize the IMM, functioning as an H^+^ pump [43]. Moreover, congenital disorders affecting the ATP synthase can impair the rotary molecular mechanism of the enzyme. causing the Δ*p* dissipation without resulting in ATP synthesis [44]. Respiratory (super)complexes have been correlated with various phenomena. in addition to their proposed roles in mitochondrial respiration. Indeed, it was discovered that the supramolecular organization of complexes dissociates when mitochondrial cristae, whose creation helps to increase respiration efficiency and whose loss is seen during regulated cell death, are disrupted [45]. However, an important role in the morphology and cristae formation is played by ATP synthase in the dimeric and oligomeric forms [46].

Indeed, *cristae*, a typical invagination of the IMM in mitochondria, serve as the focal points for OXPHOS. To manage *cristae* architecture, there are two crucial curves. According to reports, the Mitochondrial Contact site and Cristae-Organizing System (MICOS) and the ATP synthase, two key *cristae*-shaping machines, operate in an antagonistic way concerning *cristae* biogenesis [47]. The first is placed between the concave curvature of IMM at the base of the cristae folds, and the smooth expanse of the IMM is identified as the inner boundary membrane. Contrary to MICOS, which causes a negative membrane curvature, ATP synthase creates long rows of dimers that impose a positive membrane curvature (convex when viewed from the matrix, convex when viewed from the intermembrane space) [48]. Monomerization of ATP synthase dimers reveals profound changes in membrane architecture, and would affect the capacity of mitochondria to provide the cell with enough ATP to maintain vital cellular processes [49]. Mitochondrial dysfunction is linked to abnormal mitochondrial morphology that involves the dissociation of dimers involved in ATP synthase. The latter phenomenon is considered the event triggering the Mitochondrial Permeability Transition (MPT) [50].

Impaired bioenergetics is the result of mitochondrial dysfunction, and is closely related to ROS production [51]. Different conditions induce ROS production through mitochondrial polarization and cytosolic calcium (Ca^2+^) handling. Although the mitochondrial respiratory chain in the IMM is largely considered one of the major sources of ROS, other enzyme systems in mitochondria can also contribute significantly to ROS production [52]. The most significant mitochondrial superoxide generators are frequently designed as the FMN and CoQ_10_-binding sites of Complex I, and the Q_O_ site (at the outer or positive side of IMM) of Complex III_2_. In general, Complex I releases ROS into the matrix while Complex III_2_ is mostly contained in the intermembrane space [53]. Alterations in MPT, promoted by the formation and opening of Permeability Transition Pores (PTPs), may play a role in both adaptive and maladaptive reactions to oxidative stress.

Reversible PTP opening-associated ROS release, which prompts the timely release of accumulated, potentially dangerous ROS from mitochondria (joined to Ca^2+^ accumulation in the matrix), appears to be an adaptive housekeeping function. At higher ROS concentrations, longer PTP openings may result in a ROS burst that might damage mitochondria and, if it spreads from mitochondrion to mitochondrion, the cell itself. This “ROS-induced ROS release” [54] destructive activity may have a physiological function, by removing unwanted cells or damaged mitochondria. On the other hand, it can also result in the unnatural elimination of necessary mitochondria and cells.

## 4. Mitochondria Are Potentially a New Target of Natural Compounds to Counteract Inflammation

A failure of cellular homeostasis is a common accomplice of the deleterious inflammation process. Indeed, if the inflammation is not properly terminated and persists as a chronic condition it becomes pathogenic [6]. Even though the mechanisms by which dysfunctional mitochondria activate inflammatory signaling are unknown, impaired mitochondria are powerful inflammatory stimuli [4]. Nutraceuticals are bioactive compounds that promote health and have pharmaceutical potential. ROS production is a consequence of mitochondrial dysfunction and aggravates this dysfunction, triggering the inflammatory process. Nutraceuticals can counteract this, preventing abnormal inflammatory activity. The detection of and response to a wide range of pathogen- or damage-associated molecular patterns, which are generated by endogenous stress, can activate the inflammasome, a large intracellular multiprotein complex [55]. The physiological role of the downstream inflammatory pathways allows for the elimination of microbial infection and the repair of damaged tissues. The aberrant inflammation triggers pathological conditions, which are related to inflammatory disorders when dysregulated, such as aging-associated diseases, neurodegenerative diseases, diabetes, and atherosclerosis.

The primary immunological risk factors may encourage Nucleotide-binding Oligomerization Domain-like receptor (NOD) family pyrin domain containing 3 (NLRP3) inflammasome activation, wherein cells participate in interactions with TLRs. The following activation of Caspase (Cas)-1 and increased pro-inflammatory cytokine synthesis from this systemic inflammation pathway is finally caused by the maturation of interleukins and the proteolytic cleavage of Gasdermin D, which would be related to mitochondria dysfunction [56]. The latter, caused by Cas-11-dependent cleavage, localizes the N-terminal fragments to mitochondria, and forms pores, through which mtDNA is released [57]. In the cell, mtDNA has developed properties that allow it to function as a cellular sentinel for genotoxic stress [58]. Recently, it an ill-defined mechanism has been depicted, triggering the inflammasome and interferon signaling by the DNA-sensing cyclic GMP–AMP (cGAMP) synthase (cGAS), which activates Stimulator Interferon Genes (STING), an adapter protein on the Endoplasmic Reticulum (ER) membrane [59,60]. Indeed, a positive loop between the activation of the cGAS/STING pathway and the increase in ROS, promoting DNA damage, might exist. In addition to playing a critical role in cell survival and death, mitochondria have emerged as key inflammatory regulators.

The escape of mtDNA in several devastating diseases is still an obscure molecular mechanism. The critical step in the mtDNA release process is the ROS generation, which is responsible for its oxidation. Defective OXPHOS constituents or alteration of their functionality cause oxidative stress and, consequently, ROS imbalance. Antioxidants could scavenge excess ROS and reduce inflammatory responses by suppressing the inflammation phenomenon, renewing the mitochondrial bioenergetics under pathological or drug-induced imbalance conditions [61]. ROS-mediated oxidative modifications of biomolecules usually induces post-translational changes in proteins, i.e., phosphorylation, acetylation, ubiquitination, and SUMOylation, among others [62], or chemical modifications, through the oxidation of DNA bases. The most common type of DNA mutation is caused by ROS modifying guanine (G), leading to the production of 8-oxoG. This lesion causes a Hoogsteen base pair, which is a type of base pairing in nucleic acids that can result in a mismatched pairing with adenine (A) in the genome, resulting in C to A (anti) substitutions [63]. The enzymes of the base excision repair pathway are primarily responsible for the repair of 8-oxoG, achieved by exploiting their glycosylase activity. In mitochondria, 8-OxoG Glycosylase (OGG1) repairs major oxidative damage in mtDNA and, consequently, OGG1 deficiency results in increased levels of 8-oxoG in mtDNA [64]. Conversely, mitochondrial dysfunction supports Ox-mtDNA generation, in which the mtDNA misses the opportunity for OGG1 repair, and the mutated DNA is cleaved by the Flap Endonuclease-1 (FEN1). Ox-mtDNA and ROS production are not related to FEN1 activity, but instead work to respond to the creation of a bio-signal using mitochondrial Damage-Associated Molecular Patterns (mtDAMPs). Additionally, 500–650 bp Ox-mtDNA fragments are the products of mtDNA cleavage by FEN1 that exit through the pores of the mitochondrial membranes, acting as mtDAMPs (Figure 3). In the fragments of Ox-mtDNA, it was noticed that the sequences corresponding to a region within the mitochondrial genome, the D-loop, were overrepresented and released in the cytosol [65].

Ox-mtDNA is a mtDAMP and triggers the inflammatory response. Knowing the molecular process that underlies the inflammatory reactions could aid future therapeutic activity in response to microbial invasion or damage signals. Diets rich in bioactive compounds can modulate different stages of inflammation, but information related to their anti-inflammatory mechanisms is still not well understood [66]. Ox-mtDNA presence in the cytosol triggers the NLRP3 inflammasome assembly and cGAS/STING pathway-mediated type 1 interferon production [67]. Importantly, it was previously unknown as to how Ox-mtDNA moves into the cytoplasm to bind NLRP3 and also activate cGAS-STING signalling. The Voltage-Dependent Anion Channels (VDACs) and the PTP placed on the outer mitochondrial membrane (OMM) and the IMM, respectively, permit the escape of Ox-mtDNA from mitochondria to reach the cytosol, acting as a mtDAMP [4,59]. Hitherto, membrane-embedded Bax structural organization in the context of a toroidal pore of OMM [68] mediates the intrinsic apoptosis process and, after PTP-independent IMM herniation, was considered responsible for regulating DAMP signalling [69], even if robust activation of the apoptotic caspase cascade acts to suppress DAMP signaling by dying cells [70].

The spillage of Ox-mtDNA into the cytosol is allowed by the PTP opening and VDAC oligomerization (Figure 3). The first step occurs when Ox-mtDNA passes through the IMM, employing PTP with a poorly defined mechanism. PTP formation and opening are related to mitochondrial dysfunctions caused by molecular processes that abruptly increase Ca^2+^ concentrations in the matrix, as well as ROS generation [71,72,73]. Among these effects, the production of Ox-mtDNA is counted, but it is not linked to the PTP phenomenon. The second step related to Ox-mtDNA transport into the cytosol is catalyzed by VDAC oligomerization and opening that does not require Ox-mtDNA trapped in the mitochondrial intermembrane space [59]. Otherwise, the PTP and VDAC opening elicits mitochondrial depolarization, promoting Ox-mtDNA formation and its fragmentation.

The abnormal mtDNA packaging caused by ROS promotes mtDNA escape into the cytosol, where it activates the DNA sensor cGAS/STING-IRF3-dependent signalling, in order to increase interferon-stimulated gene expression and potentiate type I interferon (IFN-1) (Figure 3) [74]. Ox-mtDNA is a defective intracellular DNA detected by cGAS, and in a sequence-independent interaction, cGAS can interact with (un)self-DNA. When cGAS is activated, it converts Adenosine 5′-Triphosphate (ATP) and Guanosine 5′-Triphosphate (GTP) into cyclic GMP–AMP, known as 2′3′-cGAMP, which acts as a secondary messenger by binding and activating STING on the ER membrane. STING undergoes conformational changes that result in its translocation from the ER to the Golgi apparatus. Then, STING must be phosphorylated to be licensed for its interaction with IRF3 that, under phosphorylation posttranslational modification, dimerizes and translocates to the nucleus to drive IFN-1 expression [75] (Figure 3). Multiple molecular and cellular signalling events are related to ageing and disease, and are linked to a mitochondrial ultrastructural abnormality that occurs with inflammatory events. Recently, it was discovered that mitochondrial morphological abnormalities participated in cristae remodelling when the key proteins complexes (i.e., OPA1, MICOS, SAM, and MIB at the crista junction, whereas ATP synthase at the apex of the *cristae*) and phospholipids were altered, causing mt-DNA release and cGAS/STING pathway-dependent IFN-1 release, supporting inflammation [76]. Apart from cGAS/STING activation, responsible for the inflammatory response, the Ox-mtDNA released via PTP-VDAC channel promotes NLRP3 inflammasome, triggering its assembly in the cytosol. During the NLRP3 inflammasome activation by oligomerization mechanisms, which occur in response to an array of Ox-mtDNA fragments, the Cas-1-mediated cleavage of proinflammatory cytokines, such as interleukin 1β (IL-1β) and interleukin 18 (IL-18), into their mature active forms is triggered [55,77] (Figure 3).

The cascade reactions starting the inflammation have a common cellular event induced by ROS stimuli, generated in impaired mitochondria. Some different phytochemical compounds exploiting their antioxidant capacity have a scavenger action on ROS. As a consequence of decreasing ROS production, mitochondrial dysfunction is reduced, blocking the formation of PTP architecture and pore [78], and boosting the bioenergetic functionality of the mitochondrion [61,79]. On the balance, the antioxidant features of phytochemicals in terms of mitochondrial ROS production is translated into a cellular anti-inflammatory effect.

## 5. Phytochemicals and Phytosomes

Most plant-derived substances with biological activity (phytochemicals) are chemically identified as polyphenolic compounds, carbohydrates, lipids, terpenoids, alkaloids, other nitrogen-containing compounds and organosulfur compounds [80], as seen in Figure 4.

Among them, polyphenolic compounds (alkaloids, tannins, flavonoids, and phenolic compounds) extracted from various plant and waste matrices (citrus, Ginkgo biloba, grape seeds, hawthorn, green tea, etc.) [81,82,83], and represent the most widely used phytochemicals in pharmaceutical, cosmetic, and nutraceutical fields [84]. Several studies conducted in vitro and in vivo have shown that these compounds are endowed with multiple biological activities related more to their potent antioxidant activities. In particular, they exhibit anti-inflammatory and immunomodulatory activities [85], and are also involved in multiple mechanisms of anti-ageing activities [86]. Their potential effects are manifested in several diseases, including cancer, inflammation, neurodegenerative and cardiovascular diseases, Type 2 diabetes, and obesity, with effects on muscle atrophy and muscle health [87].

Their extraordinary potential shown in vitro is often disregarded in vivo, especially when administered orally and topically. In particular, many studies show reduced or sometimes absent action due to their poor bioavailability [88]. The chemical nature of these compounds, (large multiple-ring structures, a high molecular weight, and high polarity and miscibility in the aqueous phase), strongly conditions both their degree of liposolubility and their ability to be absorbed by passive diffusion across the lipid membranes of enterocytes [89].

All these results in vivo indicate the need to use high dosages of administration to make up for the low and erratic bioavailability of these compounds, due to poor enterocyte uptake, high pre-systemic metabolism and rapid elimination [90].

To overcome and improve both the poor absorption and low bioavailability of bioactive polyphenolic compounds, an innovative lipid-based vesicular delivery system similar to liposomes, called Phytosome, was devised and developed in the late 1980s by the company Indena (Milan, Italy) [91].

“Phyto” refers to the plant nature of the bioactive portion of the complex, and “Some” refers to the chemical characteristics of the final structure the complex takes on that is very similar to that of cell membranes [92].

The lipid phase constituents of phytosomes are used to make phenolic compounds compatible with lipids, and are represented by phospholipids obtained from soybeans, such as phosphatidylcholine, phosphatidylserine, phosphatidylethanolamine, and phosphatidylinositol [93].

Of these, the most widely used is phosphatidylcholine, as it is the main constituent of cell membranes. Structurally, it has a water-soluble head (choline) and two fat-soluble tails that make it miscible both in water and in an oily environment. Phosphatidylcholine, therefore, acts as an effective emulsifier, making the phytosome rapidly and easily absorbed in the intestinal tract. In this way, protection is promoted from the demolishing and denaturing action of digestive secretions and the intestinal bacteria, which effect the phytochemicals carried by the phytosome. In addition, the phosphatidylcholine that goes into phytosomes cannot be regarded as merely a passive “carrier” but is also a bioactive nutrient. Studies show its documented clinical efficacy for liver disease, alcoholic liver steatosis, drug-induced liver damage, and hepatitis [94].

In the study conducted by Bombardelli et al., the chemical nature of the bonds existing between the various components of phytosomes is highlighted for the first time: phospholipids and phytochemicals.

In particular, two important features of the phospholipid–phytochemical complex are highlighted: the existence of a stoichiometric ratio between the two phytosome components (1:1 or 2:1 ratio), and the formation of hydrogen bonds (H-bonds) between a molecule of phytochemicals and at least one molecule of phosphatidylcholine [95].

As shown in Figure 5, in particular, it is the nitrate and phosphate groups of choline (hydrophilic head of phosphatidylcholine) that interact with the phytochemicals, while the fat-soluble phosphatidic portion, which includes the body and tail, has the function of enveloping the choline–phytochemicals complex, generating a small microsphere, structurally similar to that of a cell. Based on the above, among all naturally occurring phytochemicals, only those that, therefore, have an active hydrogen atom (-COOH, -OH, -NH_2_, -NH, etc.), such as polyphenols, can be integrated into a phytosome structure [96].

Phytosome technology has revolutionized the pharmaceutical, nutraceutical and cosmetics industries, because phytosomes, compared with conventional plant formulations and other nano-delivery systems, have significant advantages:The phytochemical complexed in the phytosome is better protected from oxidation and degradation, improving its stability and increasing its long-term effects in pharmaceutical and cosmetic compositions [97];Compared with other nano-delivery systems, phytosomes have a simpler production method [98];The process of phytosome formation ensures efficient and meaningful drug/nutraceutical entrapment [99];Cellular uptake of the phytoactive constituents is enhanced and consequently, the required dose is reduced [100,101];Phosphatidylcholine acts both as a hepatoprotective agent, providing a synergistic effect, especially with hepatoprotective constituents found in the phytosomal complex, and as a supplement, with beneficial nutritional effects [102];Compared with liposomes, they are much smaller, and have different hydrogen bonds; the phytochemicals present in the phytosome are not dissolved in solvents or enveloped by the liposomal membrane, but are bound through chemical bonds with the polar head of phospholipids [103];Pharmacokinetic-type studies and pharmacodynamic tests conducted in experimental animals and human subjects have shown improved oral and percutaneous absorption of polar phytoactive compounds, increased bioavailability of the bioactive compound compared with non-complexed botanical derivatives, and enhanced therapeutic effects [104].The main disadvantages associated with phytosomes are:The pH sensitivity of phospholipids, due to their Zeta potential values, which must be considered when preparing phytosome formulations [105];Phytosome formation technology produces the rapid removal of phytoconstituents from the phytosome [106].

Regarding their uses and applications as shown in Figure 6, they can be used advantageously in the treatment of acute and chronic liver diseases of toxic, metabolic, infectious, or degenerative origin, in the promotion of anti-inflammatory activity, in diseases involving the cardiovascular, central and peripheral nervous, gastrointestinal, genitourinary, immune, integumentary, musculoskeletal, and respiratory systems, and in diseases concerning metabolic syndrome. Moreover, phytosomes might have a beneficial effect on electronic-cigarette smokers, since their use has been linked to an increase in redox stress/mitochondrial toxicity [107].

In the following section of this manuscript, the application of phytosomes in inflammation and antioxidant processes will be explored.

## 6. Phytosome and Pathologies

The positive effects of phytosome intake are well documented; in fact, the effects relating to the greater bioavailability and efficacy of natural compounds in some pathologies characterized by inflammatory states and excessive ROS production are known.

### 6.1. Curcumin Phytosome

It is known that human exposure to Aluminum Chloride (AlCl_3_) causes hepatotoxicity, which can be counteracted by the Curcumin Phytosome (CP). Curcumin is a polyphenolic compound extracted from *Curcuma longa*. In studies conducted in rats, Al-Kahtani et al. demonstrated that treatment with AlCl_3_ increases the concentrations of Aspartate Aminotransferase (AST), Alanine Aminotransferase (ALT), Alkaline Phosphatase (ALP), Lactate dehydrogenase (LDH), total bilirubin, and Lipid Peroxidation (LPO), reducing, in addition, the stores of albumin, Reduced Glutathione (GSH), Superoxide Dismutase (SOD) and Glutathione Peroxidase (GPx). Histological lesions have also been reported. All of this results in an increased expression of caspase-3 and a decreased expression of Bcl-2.

On the contrary, in the presence of CP, the endogenous antioxidant status is favored; therefore, caspase-3 expression is decreased and Bcl-2 expression increases, with an improvement in liver dysfunction [108]. The antioxidant and anti-inflammatory properties of curcumin and its nano-phytosome were also tested in mice that exhibited acute inflammation following the administration of carrageenan. In particular, the mice were treated, for 7 days, with an oral dose equal to 15 mg/Kg of indomethacin, curcumin and its nano-phytosome. After 7 days of treatment, the mice were administered carrageenan (1%) at the level of the subplantar region of the left paw, to induce the inflammatory process. A serum antioxidant enzyme assay found that carrageenan reduced the antioxidant activities of SOD, catalase (CAT), GPx, and glutathione reductase (GRx). Conversely, these activities increased in the presence of the curcumin nano-phytosome, both separately and in combination with indomethacin, demonstrating that curcumin nano-phytosome could enhance inflammatory-related antioxidant responses [109]. Furthermore, an oral administration of curcumin in phospholipids (Meriva^®^) has been used in subjects suffering from Chronic Kidney Disease (CKD), whose progression determines the onset of cardiovascular disease. Patients with CKD were treated for 6 months with the curcumin phytosome and assessments were made on uremic toxins, intestinal microbiota, and inflammatory and oxidative status observed during the study. Reductions of pro-inflammatory mediators such as monocyte chemoattractant protein 1 (MCP-1 or CCL-2), IFN-γ, and IL-4, as well as reductions of lipid peroxidation have been reported. Furthermore, long-term administration of the curcumin phytosome did not result in any adverse events [110].

### 6.2. Silybin Phytosome

Among the research related to the application of phytosomes, a very interesting study was conducted to improve the bioavailability of silybin, a natural compound known for its anticancer, antioxidant and hepatoprotective properties. Silybin is one of the main polyphenols found in silymarin, which is a complex of seven flavonolignans and polyphenols extracted from milk thistle (*Silybum marianum*). In particular, Chi et al. tested the phytosome-nanosuspensions formulation for silybin, defined as SPCs-NPs, in a mouse model of oxyhepatitis, induced by treatment with Carbon Tetrachloride (CCl_4_). CCl_4_ causes liver damage, following lipid peroxidation, a lower activity of antioxidant enzymes and increased generation of free radicals and ROS. The choice of silybin was dictated by the known properties of this natural compound, which is capable of protecting the liver from oxidative stress and inflammation, linked to ROS and secondary cytokine production. The CCl_4_-induced liver injury resulted in an increased presence of ALT, AST and ALP in the bloodstream, inducing centrilobular necrosis, ballooning degeneration and cellular infiltration. Only in the group of mice treated with SPCs-NPs was a significant reduction in ALT, AST and ALP levels recorded, while no significant changes in these parameters were reported in mice treated with silybin alone compared to the positive control. Furthermore, histological lesions and necrosis improved in the group treated with SPCs-NPs. This study demonstrates that the silybin phytosome increases its stability and bioavailability, resulting in the increased efficacy of silybin in hepatoprotection, allowing for more preclinical and clinical applications [111].

Furthermore, silybin, as a phytosome complex with vitamin E, was studied in a cellular model of hepatic steatosis. Indeed, steatotic rat hepatoma cells were incubated for 24 h with different doses of the silybin phytosome. Silybin reduced triglyceride accumulation by stimulating lipid catabolism and inhibiting lipogenesis. The effect of silybin on triglyceride content resulted in a reduction in ROS production and lipid peroxidation, as well as lower catalase activity and reduced NF-κB activation [112]. The silybin phytosome was also administered to rats with Cerebral Ischemia–Reperfusion (CIR) injury. Rats treated with the silybin phytosome had higher levels of SOD and glutathione and lower levels of monoaldehyde, TNF-α and IL-6 in both the hippocampus and cortex; this demonstrates the neuroprotective action of the silybin phytosome under CIR conditions [113].

### 6.3. Quercetin Phytosome

An interesting flavanol for human health is quercetin, which cannot be produced by man, but is present in abundance in foods such as fruit, especially citrus fruits, green leafy vegetables, broccoli, olive oil, cherries, and blueberries. Similar to the other flavanols, however, quercetin has a low bioavailability, due to its poor solubility in water, even if it is quite soluble in alcohol and lipids. Therefore, the quercetin-loaded phytosome nanoparticles (QP) have made it possible to overcome this difficulty, increasing the bioavailability for humans of orally administered quercetin in this formulation by about 20 times. This turns out to be very interesting, given the known antiviral, anti-atopic, pro-metabolic, and anti-inflammatory properties of quercetin. Moreover, quercetin has a protective effect on mPTP opening [114].

In particular, Di Pierro et al. collected data aimed at understanding whether the phytosomal form of quercetin could be effective in COVID-19 treatment. This idea stems from the knowledge that some of the pharmacological targets of SARS-CoV-2 are viral proteins, such as 3-Chymotrypsin-Like protease (3CLpro), Papain-Like protease (PLpro) and the Spike (S) protein, which, from molecular docking studies, were found to be inhibited by quercetin. Therefore, quercetin, in particular QP, with a higher bioavailability, could be an interesting candidate to counteract COVID-19 clinically, given its anti-inflammatory and thrombin-inhibitory properties [115]. In light of this, Di Pierro et al. investigated QP as an adjuvant therapy in 152 COVID-19 patients, to monitor the effect on the initial symptoms of the disease and the ability to prevent more serious clinical conditions. A prospective, randomized, controlled and open study was conducted, involving the administration of 1000 mg of QP per day, for thirty days. Treatment with QP reduced the frequency and duration of hospitalization, as well as the need for oxygen therapy and intensive care unit admission. The number of deaths has also decreased. Furthermore, QP appears to have anti-fatigue and pro-appetite properties. Therefore, QP as an adjuvant could have positive effects in the early stages of COVID-19 infections [116].

In addition to this, a second clinical trial, randomized, open-label and controlled, tested the effect of QP in another group of patients affected by COVID-19. In particular, for two weeks, 21 patients were treated with the Standard of Care (SC), while another 21 patients were given QP in addition to the SC. Molecular tests have shown that most of the patients also treated with QP were negative for disease within the first week of the study, presenting milder symptoms; by the second week, they all tested negative. On the contrary, the molecular tests of patients treated only with SC were still almost all positive after the first week, presenting important symptoms. The patients belonging to the latter group were all negative after about 20 days of treatment. Patients treated with QP had reduced blood levels of LDH, ferritin, CRP (C-reactive protein) and D-dimer [117].

QP has been considered as a hormone replacement therapy. This use was studied in an ovariectomized rat model, treated for 4 weeks with oral doses of QP equal to 10 and 50 mg/kg/day. The data obtained demonstrated that QP, in addition to increasing the levels of serum calcium, inorganic phosphorus and glutathione, reduced the presence of alkaline phosphatase, acid phosphatase, MDA and TNF-α. This study indicates a greater efficacy of QP in restoring a balance between ROS production and the antioxidant defense system when compared to free quercetin [118].

### 6.4. Berberine Phytosome

The efficacy of another natural product, such as berberine, was tested in women affected by Polycystic Ovary Syndrome (PCOS), an endocrine pathology characterized by hormonal imbalances, dysmetabolism and inflammation. In particular, berberine is an alkaloid used to fight infections, Type 2 diabetes and cancer, but also dyslipidemia in subjects intolerant to statins, and has been shown to enhance the expression of antioxidant enzyme activity. In this study, patients were treated with two daily oral doses of the Berberine Phytosome (BBR-PP) and evaluations were performed at baseline and after 60 days of treatment. The recorded data report a reduction in insulin resistance and acne, and an improvement in lipid metabolism and body composition, but also indicate a reduction in inflammation, with lower levels of CRP and TNF-α. Furthermore, the administration of berberine in the phytosomal formulation is characterized by good bioabsorption, which allows for the administration of lower doses, keeping the liver and kidney functions unchanged [119].

### 6.5. Mulberry and Ginger Phytosome

Phytosomes have also been used in the treatment of Metabolic Syndrome (MetS), characterized by visceral adiposity, insulin resistance, hypertension, high triglyceride levels, and low High-Density Lipoprotein Cholesterol (HDL-C) levels. Patients with MetS have an increased risk of developing Type 2 Diabetes Mellitus (T2DM) and Atherosclerotic Cardiovascular Disease (ASCVD). Both genetic and acquired factors generate oxidative stress, cellular dysfunction and the systemic inflammation process responsible for the pathogenesis of MetS [120]. In particular, the phytosome containing the combined extracts of mulberry (*Morus alba Linn. Var. Chiangmai*) and ginger (*Zingiber officinale Roscoe*) (PMG) was tested in an animal model of MetS. Specifically, male Wistar rats were fed a high-carbohydrate, high-fat diet for 16 weeks, to induce MetS. In the following 21 days, rats with MetS signs were subjected to daily oral treatment with three different doses of PMG, equal to 50, 100 and 200 mg/Kg. Data analysis demonstrated that PMG has a positive effect on body weight gain, and lipid and glucose values, as well as improving Homeostasis Model Assessment of Insulin Resistance (HOMA-IR) and Angiotensin-Converting Enzyme (ACE) levels. Even the parameters relating to the density and size of the adipocytes, to the weight of the adipose tissue, have undergone an improvement after this treatment. At the adipose tissue level, PMG also reduced inflammatory cytokines such as IL-6 and TNF-α, as well as reducing oxidative stress and Histone Deacetylase 3 (HDAC3) expression, while simultaneously increasing PPAR-γ expression. This study allows us to conclude that PMG can be considered a good adjuvant in the treatment of MetS, thanks to the content and bioavailability of phenols present in it [121].

The same authors tested the neuroprotective effect of the same doses of PMG against induced ischemic stroke in rats affected by MetS. In the presence of PMG, improvements in infarction, edema cerebral and neurological deficits were found. Under these conditions, PMG also determined a reduction of oxidative stress and inflammation, reporting a reduction of DNA-Methyltransferase 1 (DNMT 1), MDA, NF-κB, TNFα and CRP. Furthermore, SOD, CAT, GPx activity and PPARγ expression were increased [122].

### 6.6. Eufortyn^®^ Colesterolo Plus

Recently, the ANEMONE study was conducted, involving 60 healthy subjects with moderate polygenic hypercholesterolemia, treated with Eufortyn^®^ Colesterolo Plus. This nutraceutical consists of the standardized bergamot polyphenolic fraction phytosome (Vazguard^®^), as well as artichoke extract (Pycrinil^®^), artichoke dry extract (*Cynara scolymus* L.), zinc and CoQ_10_ phytosome (Ubiqosome^®^). The subjects concerned were treated for 1 month with a low-fat and low-sodium Mediterranean diet and then randomly took one pill each day of Eufortyn^®^ Colesterolo Plus or a placebo for eight weeks each. At the end of treatment, in addition to an improvement in serum lipid values, endothelial reactivity, and non-alcoholic fatty liver disease (NAFLD) indexes, an improvement in high-sensitivity C-reactive protein (hs-CRP) values was found, thus indicating a beneficial effect of this phytosome on the systemic inflammation of healthy subjects with moderate hypercholesterolemia [123].

### 6.7. Naringenin Phytosome

Furthermore, a few years ago, Yu et al. studied the effect of Naringenin (NG) on acute lung injury. This is a respiratory pathology in which the lung undergoes an important inflammatory process. NG is a plant bioflavonoid found in citrus fruits such as bergamot, grapefruit, tangerine, known for its antioxidant, anti-inflammatory, antiproliferative and anticancer properties. In this study, rats with acute lung injury were treated with Dipalmitoylphosphatidylcholine (DPPC) phytosomes NG-loaded for dry powder inhalation (NPDPIs); in particular, NPDPIs, consisting of mannitol/DPPC/NG in a 4:2:1, *w*/*w*/*w* ratio, were found to be effective. The results obtained demonstrate, first of all, that NPDPI represents a good system for the prolonged release of the drug and for improving its bioavailability in the lungs. Furthermore, a reduction of the expression of cytokines, in particular of COX-2 and Intercellular Cell Adhesion Molecule-1 (ICAM-1), and a reduction in the phosphorylation of p38 in the MAPK pathway, was recorded, indicating that NG could be responsible for the inhibition of these processes. Also, the production of ROS is reduced in the presence of NPDPIs following an increased regulation of SOD activity. Therefore, the administration of the phytosomal form of NG facilitates the action of naringenin in restoring the balance between the oxidant and antioxidant system, which is altered in acute lung injury [124].

### 6.8. Centella Asiatica Phytosome

In a mouse model of phthalic anhydride-induced Atopic Dermatitis (AD), the anti-inflammatory effect of the *Centella Asiatica* phytosome (CA phytosome) was investigated;it is a medicinal herb used in Ayurvedic, traditional African and Chinese medicine for the treatment of venous insufficiency of legs and diabetes wounds. Following the onset of AD, lesions on the dorsal skin and ear of mice were treated by applying the CA phytosome three times a week for 4 weeks. As a result, inhibition of hyperkeratosis, mast cell proliferation and inflammatory cell infiltration has been reported. The data obtained demonstrated that the CA phytosome inhibits the NF-κB signaling pathway, the release of TNF-α, IL-1β, and IgE, as well as inhibiting the expression of iNOS and COX-2 and the production of NO, resulting in it being a good treatment for AD [125].

### 6.9. Leucoselect Phytosome

A few years ago, an open-label phase I lung cancer chemoprevention study was conducted on smokers and ex-smokers to test the chemoprotective effects of the leucoselect phytosome. This phytosome consisted of Grape Seed procyanidin Extract (GSE) complexed with soy phospholipids. At the end of the first month of treatment with leucoselect phytosome, an increase in the levels of omega-3 Polyunsaturated Fatty Acids (n-3 PUFAs) was reported, in particular of eicosapentaenoic acid (EPA) and docosahexaenoic acid (DHA), known for their anticancer properties. Three months after the start of the administration of the leucoselect phytosome, an increase in the serum levels of prostaglandin E3 (PGE3) was also recorded. The latter represents a metabolite of EPA, which, in addition to having antineoplastic properties, is also an anti-inflammatory molecule. Thus, the leucoselect phytosome represents a good chemopreventive agent for lung cancer [126].

To test the immunomodulatory effect of polyphenols on inflammatory–allergic diseases, such as asthma, a study was conducted on frail elderly patients from South Italy who were administered the Leucoselect^®^ Phytosome^®^. The latter is a dietary supplement enriched in the flavonoid epigallocatechin. The results report the upregulation of the T Helper (Th)1 response by the Leucoselect^®^ Phytosome^®^, with a higher concentration of IL-2 and interferon-γ in serum. In addition, this treatment produced a balance between IL-17, with its pro-inflammatory action, and IL-10, with its anti-inflammatory role [127].

### 6.10. CoQ10 Phytosome

Moreover, in humans, the effect of CoQ_10_ deficiency is known. CoQ_10_ is a natural compound known for its antioxidant properties, its ability to prevent damage caused by free radicals, as well as its ability to inhibit inflammation signaling pathways. The CoQ_10_ deficiency can be of a primary or secondary form, and determines the onset of serious diseases, such as encephalomyopathy, cerebellar ataxia and cardiovascular disease. Therefore, maintaining correct concentrations of CoQ_10_ is important from a therapeutic point of view, even if it is very difficult to achieve this result given its high molecular weight, high lipophilicity, light sensitivity and thermolability [128]. To solve these problems, over the years, studies have been conducted on various formulations of CoQ_10_ that could increase its oral bioavailability. One such study is by Rizzardi et al., published in 2021. In this in vitro study, the authors evaluated the effects of the CoQ_10_ phytosomal formulation (UBIQSOME, UBQ) on the bioenergetic and antioxidant status of human intestinal epithelial cells (I407) and rat cardiomyoblasts (H9c2). In this formulation, CoQ_10_ was administered in association with lecithin. The positive results obtained indicate a higher concentration of CoQ_10_ both at the cellular and mitochondrial levels, and a different distribution of CoQ_10_ in the two cell lines, which indicates a limitation in the absorption at the intestinal level. UBQ increased both the reduced and oxidized forms of CoQ_10_ in I407 cells, protecting them from oxidative stress, while in H9c2 cells it increased the oxidized form more, generating less protection against oxidative stress. However, this data demonstrated that the antioxidant effect of CoQ_10_ is linked to the ability of cells to reduce it, and not to its cellular bioavailability. In fact, against oxidative stress, a formulation that increases bioavailability seems more effective, allowing incubation at lower doses, thus preventing a non-specific accumulation of intracellular CoQ_10_. This result seems to have been achieved with this UBQ formulation, in addition to obtaining a lower lipid peroxidation of cell membranes and protection from ferroptosis, which represents a type of cell death associated with various neurological diseases. UBQ favorably increased the FCCP/oligomycin A oxygen consumption ratio, increasing spare capacity, which favors energy requirements. The UBQ phytosome increased cellular ATP and protein content, mitochondrial transmembrane potential, and citrate synthase activity. The latter is a mitochondrial mass marker that could indicate an increase in mitochondrial biogenesis, whose function is to protect against oxidative stress [129].

A double-blind, randomized, placebo-controlled, crossover clinical study was conducted on 20 healthy young non-smoking subjects to test the CoQ_10_ phytosome. In the clinical study by Cicero et al., the effect of the CoQ_10_ phytosome on the endothelial reactivity and the plasma Total Antioxidant Capacity (TAC) was tested, administered in a single dose of 150 mg or a double daily dose, depending on whether the integration was done chronically or acutely. In both cases, the CoQ_10_ phytosome improved plasma TAC and endothelial reactivity, the dysfunction of which is due to increased ROS caused by NO deficiency, with increased vascular smooth muscle growth within the endothelium, resulting in the onset of the atherosclerotic phenomenon [128].

### 6.11. Phytosomes and Cytotoxicity

Much evidence points to the efficacy of phytosomal formulations in increasing the cytotoxicity of certain natural compounds. Specifically, Alhakamy et al. conducted a study on treating ovarian cancer cells, OVCAR-3, with the phytosome Icariin (ICA). ICA is a flavonol glycoside found in *Epimedium* grandiflorum. It is best known for its efficacy in treating atherosclerosis and neurodegenerative disorders. It also possesses antioxidant, anti-inflammatory, cardioprotective and hepatoprotective activities. In addition, the antitumor activity of ICA has been demonstrated through its cytotoxicity, apoptotic activity, and regulation of cell cycle protein expression against different cell types. ICA also has anti-angiogenic, anti-metastatic, and immunomodulatory effects, explicitly enhancing the chemosensitivity of ovarian cancer cells. The limitation of ICA treatments is related to its pharmacokinetic properties; it has poor bioavailability due to its chemical structure and a short half-life when administered orally (3.15 h) and intravenously (0.56 h). Therefore, the phytosomal formulation seems to have many advantages for the cytotoxic efficacy of ICAs, particularly for ovarian cancer. This pathology represents a very serious and widespread gynecologic neoplasm worldwide, being fatal in most cases, as it has no early symptoms or adequate treatment. In the study by Alhakamy et al. an increase in the cytotoxic activity of the ICA phytosome on OVCAR-3 cells was demonstrated. Specifically, there was an increase in ICA phytosome-treated cells in the G2/M and pre-G1 phases of the cell cycle. Furthermore, Annexin V staining demonstrated an increase in early, late, and total apoptotic cells. In addition, ICA phytosome treatment impaired Mitochondrial Membrane Potential (MMP), significantly increased caspase-3 content, and increased ROS generation. Specifically, MMP was determined using the Tetramethyl Rhodamine Methyl ester (TMRM) assay kit as a probe. The reported results record a 15% reduction in MMP in the presence of the ICA phytosome. All this indicates an increased cytotoxic effect of ICA, in the phytosomal formulation, on ovarian cancer cells [130]. The increased cytotoxic activity of phytochemicals in the phytosomal formulation was also demonstrated by the research of Al-Rabia et al. These researchers studied the antitumor effect of the Curcumin (CUR) phytosome conjugated to Scorpion Venom (SV) in PC3 human prostate cancer cells. This cancer is prevalent and high in incidence, especially in Europe, North America and Australia. It is characterized, in particular, by altered androgen receptor function. However, the drugs used to treat this pathology have a high toxicity. Therefore, approaches that may be effective but less toxic are being investigated. For example, research has shown that Curcumin–Phospholipon from Scorpion Venom (CUR–PL-SV) has a higher cytotoxicity and a lower IC_50_ value than curcumin and PL-SV used individually. In addition, treatment with CUR–PL-SV resulted in cell cycle arrest in the G2-M and pre-G1 phases. Furthermore, Annexin V staining revealed an increase in early and late apoptotic PC3 cells and the number of necrotic cells. This study also revealed that treatment with CUR–PL-SV reduced mitochondrial membrane permeability, decreased the levels of Bcl-2, NF-kB, and TNF-α, and increased the levels of Bax, caspase-3, and p53, indicating a greater antitumor capacity of curcumin in the phytosomal formulation [131]. Neamatallah et al. conducted a study on Andrographolide (AG) to verify that a phytochemical compound increases its antitumor activity when used in a phytosomal formulation. AG, a bicyclic diterpenoid lactone isolated from *Andrographis paniculata*, is effective against renal, lung, ovarian, colon, liver, and prostate cancers. This molecule was tested as AG-loaded Phytosomes (AG-PTM) in the liver cancer cell line HepG2. This type of cancer has a very high mortality rate. Usually, in the early stages, patients are asymptomatic; unfortunately, symptoms appear only in the more advanced stages of the disease. In addition, available therapies can cause resistance and severe side effects. Therefore, possible therapies with phytochemicals could be helpful and interesting in these cases. In the study by Neamatallah et al., HepG2 cells showed an increased uptake of AG in the phytosomal formulation, resulting in cell cycle arrest in the G2-M phase and increased mitochondrial-dependent apoptosis. Indeed, increased ROS production and decreased MMPs were recorded, as well as an increased expression of Bax and caspase-3 and decreased expression of Bcl-2 [132].

## 7. Conclusions

In recent years, numerous studies have determined the importance of using phytosomes as a preventive treatment or an adjunct to drug therapies. This is because using phytosomes increases the bioavailability and efficacy of natural compounds, which have long been considered effective non-pharmacological therapies. Clinical studies, in vivo and in vitro, demonstrate the positive effects of phytosome supplementation on the onset and management of many diseases that are characterized by an inflammatory state or excessive ROS production. It is well documented that phytochemicals in the phytosomal formulation can promote the expression of Bcl-2 and reduce that of caspase-3, while also increasing the antioxidant activities of SOD, glutathione, CAT, GPx, and GRx, enhancing inflammation-related antioxidant responses. In addition, phytosomes promote the reduction of pro-inflammatory mediators such as MCP-1, IFN-γ, TNF-α, MDA, IL-4 and IL-6, and CRP, and of lipid peroxidation and NF -κB activation, as well as inhibition of iNOS and COX-2 expression and NO production. In addition, at the mitochondrial level, phytosomal formulations can promote cellular ATP and protein content, transmembrane potential, and citrate synthase activity. In addition, at the level of cancer cells, they amplify the cytotoxicity of the phytochemicals with which they are conjugated, as they promote the impairment of MMPs, increase caspase-3 content, and result in increased ROS production and thus trigger more apoptotic events in cancer cells. Phytosomes amplify the antioxidant and anti-inflammatory effects of natural compounds by increasing their stability, bioabsorption, and bioavailability. In this way, desired results can be achieved at lower dosages and for extended periods, allowing for greater preclinical and clinical applications. This action makes phytosomes hepatoprotective and neuroprotective anti-atherosclerotic agents, and they can provide benefits in cancer and COVID-19 patients. In light of this, phytosomes could be another tool to more selectively and effectively counter inflammation and oxidative stress at the mitochondrial level. This review highlights the close correlation between inflammation, ROS production, mitochondria, and the positive therapeutic effects of phytosomes, in order to plan future research for developing targeted therapies.

## Figures and Tables

**Figure 1 ijms-24-06106-f001:**
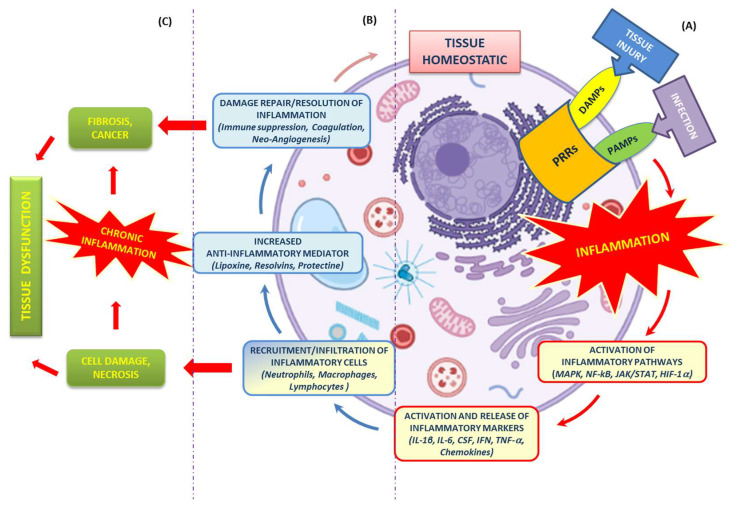
Mechanisms of the inflammatory response. (**A**) Tissue homeostasis can be altered by external stimuli (tissue injury or infection), resulting in the activation of the innate immune system (PRRs). Such activation triggers an inflammatory response cascade (red arrows) driven by multiple pathways (MAPK, NF-kB, JAK/STAT, HIF-1α) and the release of inflammatory markers (IL-1β, IL-6, CSF, IFN, TNF-, chemokines) at the site of injury. (**B**) This release is aimed at the recruitment and infiltration of various immune cells (neutrophils, macrophages and lymphocytes). Their presence ensures both the continuation of “transient” inflammation, which is useful for eliminating the cause of tissue injury, as well as the secretion of anti-inflammatory and pro-regenerative cytokines that promote the resolution of inflammation, tissue repair and restoration of tissue homeostasis (blue and pink arrows). (**C**) Uncontrolled or chronic inflammation, promoted by cellular damage/necrosis, generates remodeling (fibrosis and cancer) and tissue dysfunction.

**Figure 2 ijms-24-06106-f002:**
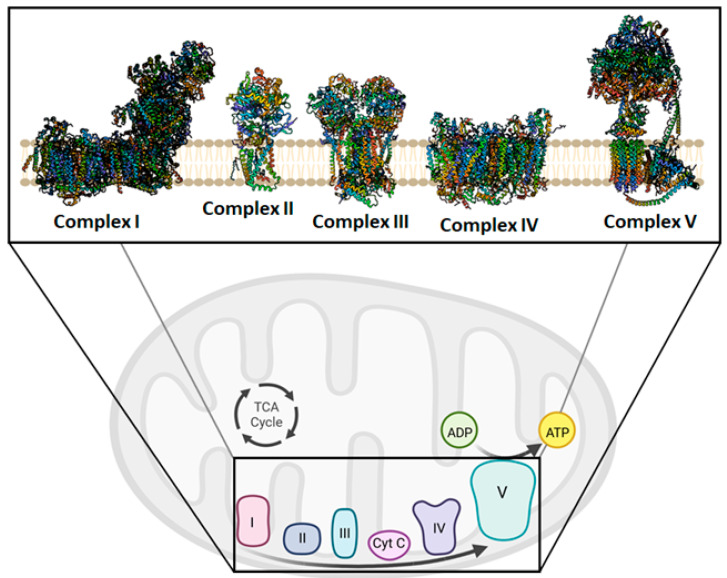
Overview of the mitochondrial oxidative phosphorylation. In the insert, Complex I, II, III, IV, and V are shown in their free form drawn as ribbon representations, obtained from modified PDB IDs: 6YJ4, 1ZOY, 6FO0, 7CP5, and 6TT7, respectively. This figure was created using BioRender (17 December 2022; https://biorender.com/).

**Figure 3 ijms-24-06106-f003:**
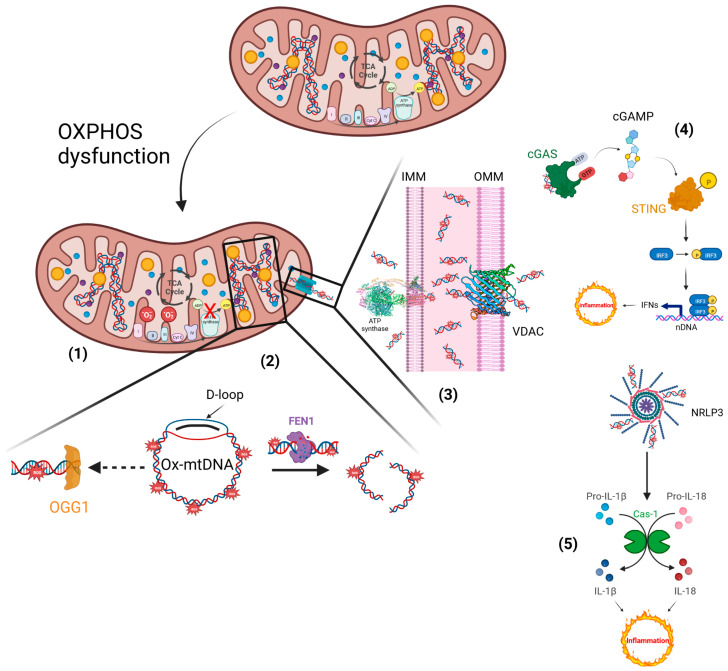
Inflammatory responses elicited by impaired mitochondria. (1) Defective electron transfer in OXPHOS produces ROS, responsible for the oxidation of mtDNA. (2) Ox-mtDNA might be repaired by OGG1 or cleaved by FEN1 to fragments that are expelled from mitochondria. (3) PTP and VDAC are located on the IMM and OMM, respectively, and translocate the Ox-mtDNA into the cytosol to activate the inflammatory reactions. (4) OX-mtDNA acts as mtDAMPs and promotes cGAS/STING signalling, responsible for type I IFN production. (5) In addition, OX-mtDNA binds NRLP3 to trigger inflammasome assembly and caspase 1 (Cas-1) activation, and consequently promotes proteolytic maturation of the biologically inactive precursors of interleukin-1β (IL-1β) and interleukin-18 (IL-18), thus generating potent proinflammatory and pyrogenic activities. This figure was created using BioRender (7 March 2023; https://biorender.com/).

**Figure 4 ijms-24-06106-f004:**
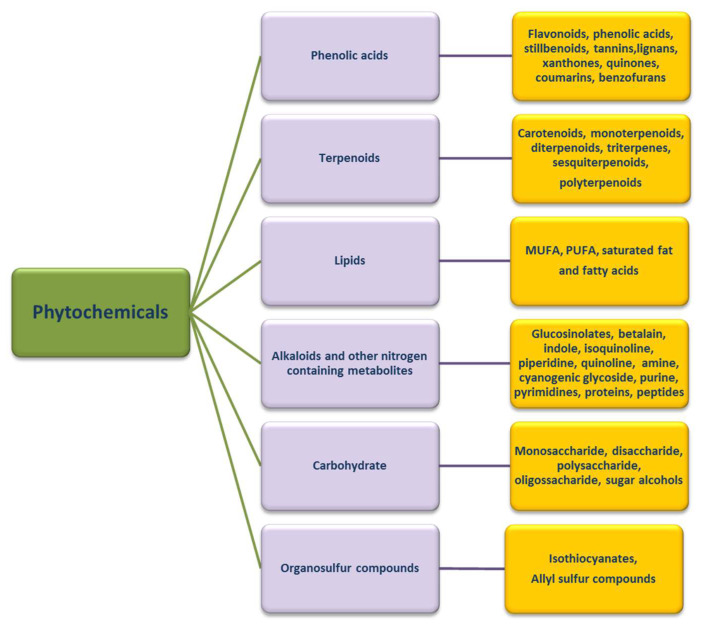
Categories of phytochemicals. There are six main categories of phytochemicals (phenolic acids, terpenoids, lipids, etc.), distinguished according to their chemical structure. Each category can be subdivided into subgroups based on the functional groups that characterize their molecule, and confer particular and specific biological properties.

**Figure 5 ijms-24-06106-f005:**
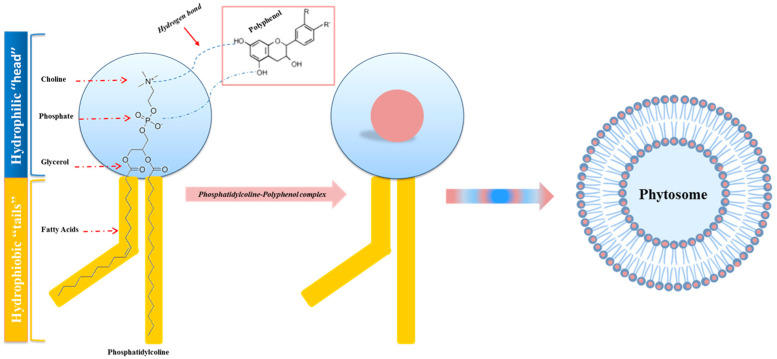
Process of phytosome formation. Schematic and structural representation of the hydro-gen bonding (dashed line) between the phytochemical (polyphenols) and the hydrophilic “head” of the phospholipid (phosphatidylcholine), which generates the phosphatidylcholine–polyphenol complex. The hydrophobic “tails”, comprising the body and tail, envelop the phosphatidylcholine–polyphenol complex, causing a small microsphere structurally similar to a cell: the phytosome.

**Figure 6 ijms-24-06106-f006:**
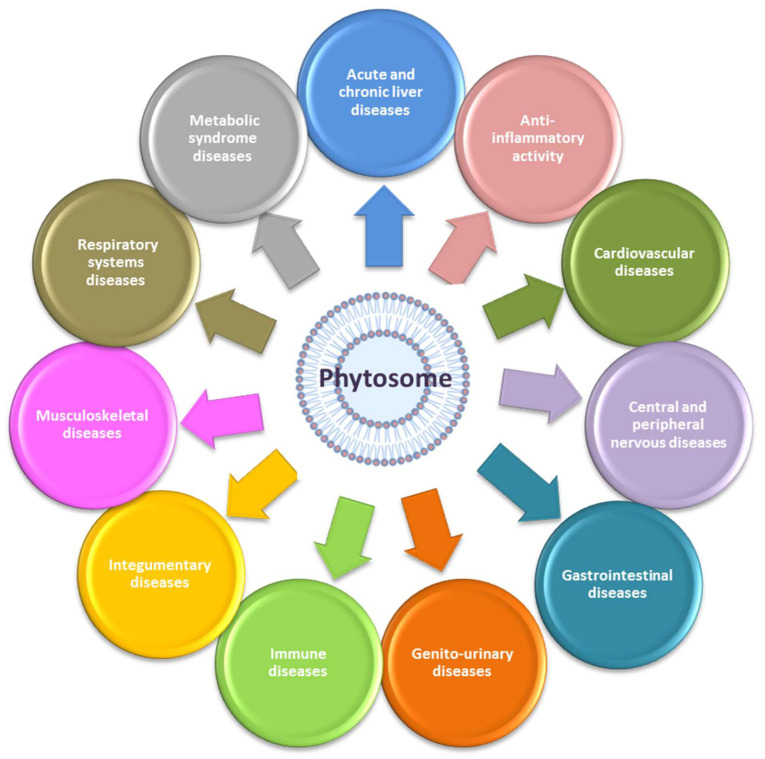
Phytosomes and their biological activities. Phytosomes have a wide variety of beneficial effects on multiple diseases involving most of the systems of the human body. The most representative uses are in the treatment of acute and chronic liver diseases of toxic, metabolic, infectious, or degenerative origin; in promoting anti-inflammatory activity; in diseases involving the cardiovascular, central and peripheral nervous, gastrointestinal, genitourinary, immune, integumentary, musculoskeletal, and respiratory systems; in diseases concerning the metabolic syndrome.

## Data Availability

Not applicable.

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
