# Peer review of "Inflammation, Mitochondria and Natural Compounds Together in the Circle of Trust"

_ijms, 2023, doi:10.3390/ijms24076106_

Round 1

Reviewer 1 Report

The authors addressed the relationship between mitochondrial dysfunction, inflammation, and the beneficial effects of natural compounds. The inflammatory cellular response to mitochondria DAMP is a hot topic in research and the data presented is useful for readers and experts in the field. Nevertheless, the text would need some improvements as the ideas are not clearly described. The present review is too extensive and difficult to follow particularly section 5 “Phytosome and pathologies” which is too long.

I have some suggestions/comments for the authors:

- Figures 1 and 3 could benefit from adding numbers to the image and describing them accordingly in the legends.

- The legends from Figures 4, 5, and 6 should be more descriptive for easier interpretation.

 - Line 308 “Aberrant mitochondrial morphology is joined to mitochondrial dysfunction and the dissociation of ATP synthase is also considered the event triggering the mitochondrial phenomenon inducing mitochondrial permeability transition (MPT) [57]." This sentence is not clear.

-Line 314 "The mitochondrial respiratory chain in the IMM is usually considered one of the major sources. Although the mitochondrial respiratory chain in the IMM is largely considered one of the major sources of ROS, other enzyme systems in mitochondria can also contribute significantly to ROS production [59]." The two sentences repeat themselves.

 - Line 340 “Mitochondrial dysfunction, such as the production of ROS, is one of several molecular and cellular events that nutraceuticals can counteract to prevent abnormal inflammation activation”

ROS production is a consequence of mitochondrial dysfunction and aggravates the dysfunction. It is not clear what the authors meant.

 - Line 688 “Phytosomes have also been used in the treatment of metabolic syndrome (MetS).”

A brief description of the metabolic syndrome should be included.

Author Response

The authors addressed the relationship between mitochondrial dysfunction, inflammation, and the beneficial effects of natural compounds. The inflammatory cellular response to mitochondria DAMP is a hot topic in research and the data presented is useful for readers and experts in the field. Nevertheless, the text would need some improvements as the ideas are not clearly described. The present review is too extensive and difficult to follow particularly section 5 “Phytosome and pathologies” which is too long.

We have followed the kind suggestion of the Reviewer and have divided section 5 “Phytosome and pathologies” into sub-paragraphs

I have some suggestions/comments for the authors:

- Figures 1 and 3 could benefit from adding numbers to the image and describing them accordingly in the legends.

We thank the reviewer for the suggestion: we have supplemented figures 1 and 3 with numbers and updated the captions.

- The legends from Figures 4, 5, and 6 should be more descriptive for easier interpretation.

We thank the reviewer for the suggestion: we have made the figure captions, 4,5,6, more descriptive and easy to interpret.

 - Line 308 “Aberrant mitochondrial morphology is joined to mitochondrial dysfunction and the dissociation of ATP synthase is also considered the event triggering the mitochondrial phenomenon inducing mitochondrial permeability transition (MPT) [57]." This sentence is not clear.

Thank you for pointing out this deficiency. We have rewritten the sentence.

-Line 314 "The mitochondrial respiratory chain in the IMM is usually considered one of the major sources. Although the mitochondrial respiratory chain in the IMM is largely considered one of the major sources of ROS, other enzyme systems in mitochondria can also contribute significantly to ROS production [59]." The two sentences repeat themselves.

We thank the Reviewer for spotting this mistake. We have edited the text.

 - Line 340 “Mitochondrial dysfunction, such as the production of ROS, is one of several molecular and cellular events that nutraceuticals can counteract to prevent abnormal inflammation activation” ROS production is a consequence of mitochondrial dysfunction and aggravates the dysfunction. It is not clear what the authors meant.

We have rephrased the sentence, thank you.

 - Line 688 “Phytosomes have also been used in the treatment of metabolic syndrome (MetS).” A brief description of the metabolic syndrome should be included.

In subsection 5.5 "Mulberry and ginger phytosome" we have added a brief description of the metabolic syndrome

Reviewer 2 Report

In this review Nesci et al. have provided a pleasant summary of current knowledge and recent advances made in understanding the role of phytosomes in inflammatory-driven diseases, in particular mitochondrial dysfunction. Furthermore, the review specifically focuses on reactive oxygen species (ROS)-mediated inflammation and the potential of phytosomes to be used in combination with alkaloids, polyphenols and coenzyme Q10 to scavenge ROS and alleviate inflammation.

This review is well written and structured and covers important aspects of mitochondrial function, mitochondrial-mediated inflammation and phytochemicals. While the review is well laid out and systematically covers various aspects of mitochondrial-driven inflammation, some parts of the review require additional work (please see specific comments below).

1.    The review starts off with a general introduction about “inflammation”. While I appreciate the authors’ effort, inflammation is a vast topic with multiple pathways that are difficult to summarize in a few paragraphs. It would benefit the authors to shorten this part of the review and cite other review articles where inflammation has been covered extensively rather than try to summarize it here. Instead, the authors should focus their attention more on mitochondrial-driven inflammation (see point 2 below)

2.    Since the authors main goal is to focus mitochondrial-mediated inflammation, this part demands more details in the review. The authors summarize the ROS-mediated aspect of mtDNA inflammation however there are other aspects of mitochondrial-driven inflammation that are not covered. For example, oxidized proteins and lipids are not represented fully in the review. Importantly, mtRNA has not been mentioned at all. It would benefit the authors (and readers) to have sections on these topics, including figures.

3.    In the last part of the review the authors discuss extensively about phytochemicals and phytosomes. The literature cited so far seems like a summary of all that is known since 2014 until today about the role of phytosomes in curbing inflammation. However, specific sections within this part of the review detailing out the roles of these phytosomes in specific diseases is missing. Most importantly, a section specifically describing the studies so far on the use of phytosomes (or phytochemicals) in combatting mitochondrial diseases and mitochondrial-driven inflammation is missing.

4.    Finally, a section (or paragraph) on how the mechanism of action of these phytosomes can be used/manipulated in curbing inflammation is lacking. Do these phytochemicals improve antioxidative capacity of all cells thus improving inflammation? Can this be made into a more targeted approach towards therapy?

Author Response

In this review Nesci et al. have provided a pleasant summary of current knowledge and recent advances made in understanding the role of phytosomes in inflammatory-driven diseases, in particular mitochondrial dysfunction. Furthermore, the review specifically focuses on reactive oxygen species (ROS)-mediated inflammation and the potential of phytosomes to be used in combination with alkaloids, polyphenols and coenzyme Q10 to scavenge ROS and alleviate inflammation.

This review is well written and structured and covers important aspects of mitochondrial function, mitochondrial-mediated inflammation and phytochemicals. While the review is well laid out and systematically covers various aspects of mitochondrial-driven inflammation, some parts of the review require additional work (please see specific comments below).

  1. The review starts off with a general introduction about “inflammation”. While I appreciate the authors’ effort, inflammation is a vast topic with multiple pathways that are difficult to summarize in a few paragraphs. It would benefit the authors to shorten this part of the review and cite other review articles where inflammation has been covered extensively rather than try to summarize it here. Instead, the authors should focus their attention more on mitochondrial-driven inflammation (see point 2 below)

We thank the reviewer for the valuable suggestion: we have shortened the paragraph about inflammation. We have removed some parts and added new citations, in which these topics are extensively described

  1. Since the authors main goal is to focus mitochondrial-mediated inflammation, this part demands more details in the review. The authors summarize the ROS-mediated aspect of mtDNA inflammation however there are other aspects of mitochondrial-driven inflammation that are not covered. For example, oxidized proteins and lipids are not represented fully in the review. Importantly, mtRNA has not been mentioned at all. It would benefit the authors (and readers) to have sections on these topics, including figures.

Nowadays it is proposed that the main event of mitochondrial dysfunction with the generation of ROS is related to inflammatory processes with mtDNA oxidation. Therefore, in this review, we have reviewed the latest research on this topic. Oxidation of cysteines and amino acids with aromatic rings as well as lipid peroxidation is not a specific event for inflammation with impaired mitochondria. However, we are willing to deepen this topic proposed by the Reviewer if could provide us with bibliography references. Regarding the ass mtRNA-inflammation, we are willing to add this information if the Reviewer has references to suggest.

  1. In the last part of the review the authors discuss extensively about phytochemicals and phytosomes. The literature cited so far seems like a summary of all that is known since 2014 until today about the role of phytosomes in curbing inflammation. However, specific sections within this part of the review detailing out the roles of these phytosomes in specific diseases is missing. Most importantly, a section specifically describing the studies so far on the use of phytosomes (or phytochemicals) in combatting mitochondrial diseases and mitochondrial-driven inflammation is missing.

As kindly suggested by the reviewer we have added section 5.11 "Phytosomes and cytotoxicity" to describe the role of phytosomes in cancer cells where phytochemicals have a specific mitochondrial effect.

  1. Finally, a section (or paragraph) on how the mechanism of action of these phytosomes can be used/manipulated in curbing inflammation is lacking. Do these phytochemicals improve antioxidative capacity of all cells thus improving inflammation? Can this be made into a more targeted approach towards therapy?

Thanks, done. As suggested by the reviewer we have modified section 6 "Conclusions" to better describe the role of phytosomes in enhancing the anti-inflammatory and antioxidant properties of phytochemicals for future research that may improve the therapeutic application of phytosomes.

Round 2

Reviewer 1 Report

The authors have addressed all the raised concerns, turning the manuscript more clear and easy to follow. I have no further comments and recommend the acceptance in the current form.

Reviewer 2 Report

Thank you for all the additions to the review